

# Influence of conservation tillage on Greenhouse gas fluxes and crop productivity in spring-wheat agroecosystems on the Loess Plateau of China

Abdul-Rauf Malimanga Alhassan[1,2,*], Chuanjie Yang[1,*], Weiwei Ma[1] and Guang Li[1]

[1] College of Forestry, Gansu Agricultural University, Lanzhou, Gansu, China
[2] Department of Water Resources and Sustainable Development, University of Environment and Sustainable Development, Somanya, Eastern Region, Ghana
[*] These authors contributed equally to this work.

Corresponding author
Guang Li, liguang468@yahoo.com

## ABSTRACT

The effects of climate change such as dry spells, floods and erosion heavily impact agriculture especially smallholder systems on the Northwestern Loess Plateau of China. Nonetheless agriculture also contributes to global warming through the emission of greenhouse gases such as $CO_2$, $CH_4$ and $N_2O$. Yet this complex conundrum can be alleviated and mitigated through sound soil and water management practices. Despite considerable literature on Conservation Agriculture (CA) as a strategy to improve the resilience and mitigation capacity of agroecosystems, there is still paucity of information on the impacts of CA on crop production and environmental quality on the Plateau. In order to fill this gap this study examined the effects of no-till and straw mulch on crop productivity and greenhouse gas fluxes in agroecosystems on the Plateau where farmers' common practice of conventional tillage (CT) was tested against three CA practices: conventional tillage with straw mulch (CTS), no-till (NT) and no-till with straw mulch (NTS). The results indicated that all three CA practices (CTS, NT and NTS) markedly increased soil water content (SWC), soil organic carbon (SOC) and soil total nitrogen (STN) but reduced soil temperature (ST). Average grain yields were 854.46 ± 76.51, 699.30 ± 133.52 and 908.18±38.64 kg ha$^{-1}$ respectively under CTS, NT and NTS indicating an increase by approximately 33%, 9% and 41% respectively compared with CT (644.61 ± 76.98 kg ha$^{-1}$). There were significant ($p < 0.05$) reductions of Net $CO_2$ emissions under NT (7.37 ± 0.89 tCO2e ha$^{-1}$y$^{-1}$) and NTS (6.65 ± 0.73 tCO2e ha$^{-1}$y$^{-1}$) compared with CTS (10.65 ± 0.18 tCO2e ha$^{-1}$y$^{-1}$) and CT (11.14 ± 0.58 tCO2e ha$^{-1}$y$^{-1}$). All the treatments served as sinks of $CH_4$ but NTS had the highest absorption capacity (−0.27 ± 0.024 tCO2e ha$^{-1}$y$^{-1}$) and increased absorption significantly ($p < 0.05$) compared with CT (−0.21 ± 0.017 tCO2e ha$^{-1}$y$^{-1}$); however, CA did not reduce emissions of $N_2O$. These had an influence on Global warming potential (GWP) as NT and NTS resulted in significant reduction in net GWP. Grain yield was significantly correlated positively with SOC and STN ($p < 0.05$); ecosystem respiration was also significantly correlated with SWC and ST while $CH_4$ flux was highly correlated with ST ($p < 0.001$). Crop yield and GHG responses to CA were controlled

![PeerJ]

by soil hydrothermal and nutrient changes, thus improving these conditions through adoption of sustainable soil moisture improvement practices such as no-till, straw mulch, green manuring, contour ploughing and terracing can improve crop resilience to climate change and reduce GHG emissions in arid and semi-arid regions.

## INTRODUCTION

Agricultural soils are potential sources of carbon dioxide ($CO_2$), nitrous oxide ($N_2O$) and methane ($CH_4$) (*Smith et al., 2008*). These gases constitute the most important greenhouse gases (GHGs) and their emissions from agriculture and land-use change account for one-third of global warming (*Cole et al., 1997*). Meanwhile agriculture is also one of the most affected sectors by climate change through several climate induced processes. Changes in hydrological cycle and temperature affects crop cultivation in various ways: higher temperatures may cause shortening of the crop cycle in arid and semi-arid areas, resulting in low yields (*IPCC, 2007*) while lower precipitation may cause moisture deficit under rainfed cultivation, which could also result in significant yield decline (*Calzadilla et al., 2013*). In all these complexities, agriculture still holds a potential to adapt to climate change through sound management practices and as well reduce its contribution to global warming through carbon sequestration and less GHG emissions.

Wheat is a crop with global importance (*Huang et al., 2003*) and is central to global food security. Its cultivation in China occupied approximately 24 million hectares (*Li et al., 2019*). On the Loess Plateau region of Western China, wheat accounts for 35% of the region's total production area and 40% of total crop production volumes (*An et al., 2014*). The Loess Plateau, however, is a fragile dryland area with abundant smallholder farmers whose activities are threatened by wind and water erosion. Coupled with wide adoption of rainfed agriculture and conventional tillage (CT) practices, the resilience of production systems to climate change is threatened by the intricate interaction of environmental and anthropogenic factors, increasing the risks of farmers to food insecurity and poor livelihood. Innovative soil management practices hold huge potential in alleviating the effect of climate change on production systems and vice versa.

Tillage, though an important component of crop cultivation may affect soil carbon (C) cycle. The practice of CT where mechanical means is employed in land preparation causes rapid soil organic matter decomposition and oxidation of soil C to $CO_2$ (*Reicosky, 1997*; *Six, Elliott & Paustian, 2000*). This may affect changes in soil structure which could influence soil water holding capacity, soil fertility and GHG emissions. Under conservation tillage soil disturbance is minimal which maintains soil physico-chemical and biological properties, thereby improving soil water storage capacity. In addition, the provision of soil cover or amendments increases soil organic matter and nutrient content which may enhance crop yield. Crop yield is dependent on soil suitability and limited by soil physical

properties (*Indoria et al., 2016*), chemical properties (*Wang et al., 2008*) and biological properties (*Woźniak & Gos, 2014*). Management practices that would facilitate meeting global food demand and conserving the already stressed environment (*Lal, 2005*) is key to sustainable crop production. No-till with residue retention is a key conservation agriculture (CA) practice that has been reported to improve soil condition (*Li et al., 2014*), increased rainfed crop yield (*Pittelkow et al., 2015*) and increased soil C stocks (*Paustian et al., 2006*). But these responses to conservation tillage is variable in literature with reports of increased yields (*Fabrizzi et al., 2005*), reduced yields (*Taa, Tanner & Bennie, 2004*) and no effect (*Lampurlanés, Angas & Cantero-Martinez, 2002*). Different responses are dependent on several factors such as environment, duration of implementation and types of conservation practices adopted (*Zheng et al., 2014*). It is not clear how the drylands of the Loess plateau will respond to conservation tillage. Furthermore, studies on GHG response to conservation tillage on the Loess Plateau are scarce and much is still unknown. This research is needed in order to provide tailor-made recommendations for sustainable and climate-smart crop production on the plateau.

Thus the objective of this study was (1) to examine the influence of no-till and straw mulching soil management practices on crop yield (2) to analyse the dynamics of $CO_2$, $CH_4$ and $N_2O$ fluxes as affected by conservation tillage and (3) to identify the mechanisms that control the responses of yield and greenhouse gases to tillage practices in dryland areas.

## MATERIALS & METHODS

### Study area

This experiment was conducted for two years (2017-2018) in the Anjiapo catchment on the western Loess Plateau in Gansu province at the Soil and Water Conservation Research Institute in Dingxi (35°34′53″N, 104°38′30″E; 2,000 m above sea level). For this study we have continuous data of forty two years (precipitation-385 mm, evaporation-1531 mm, sunshine duration-2448 h, temperature-7.1 °C, and frost free period-153 days). The soil is formed from Loess with a sandy-loam texture, with average soil bulk density of 1.26 $gcm^{-3}$. Average soil organic carbon (SOC) was 6.21 $gkg^{-1}$ while total nitrogen content was 0.61 $gkg^{-1}$. Precipitation, maximum and minimum temperatures for the experimental period are shown in Fig. 1.

### Experimental design

Four tillage treatments were established in a randomized complete block design. The treatments included conventional tillage (CT), conventional tillage with straw mulch (CTS), no-till (NT) and no-till with straw mulch (NTS). Sowing was conducted in spring (mid-March) in both years while crops were harvested in late July to early August. In the tilled plots, soils were tilled at two different times by manual inversion with shovels to a depth of 20 cm; first in October of the previous year and again in March just before planting. Glyphosate (30%) herbicide was applied to control weeds in the plots. Wheat straw (dry weight of 3.75 ton/ha) was spread uniformly on all straw-treated plots immediately after planting. Chemical composition of the wheat straw is shown in Table 1. Planting was done

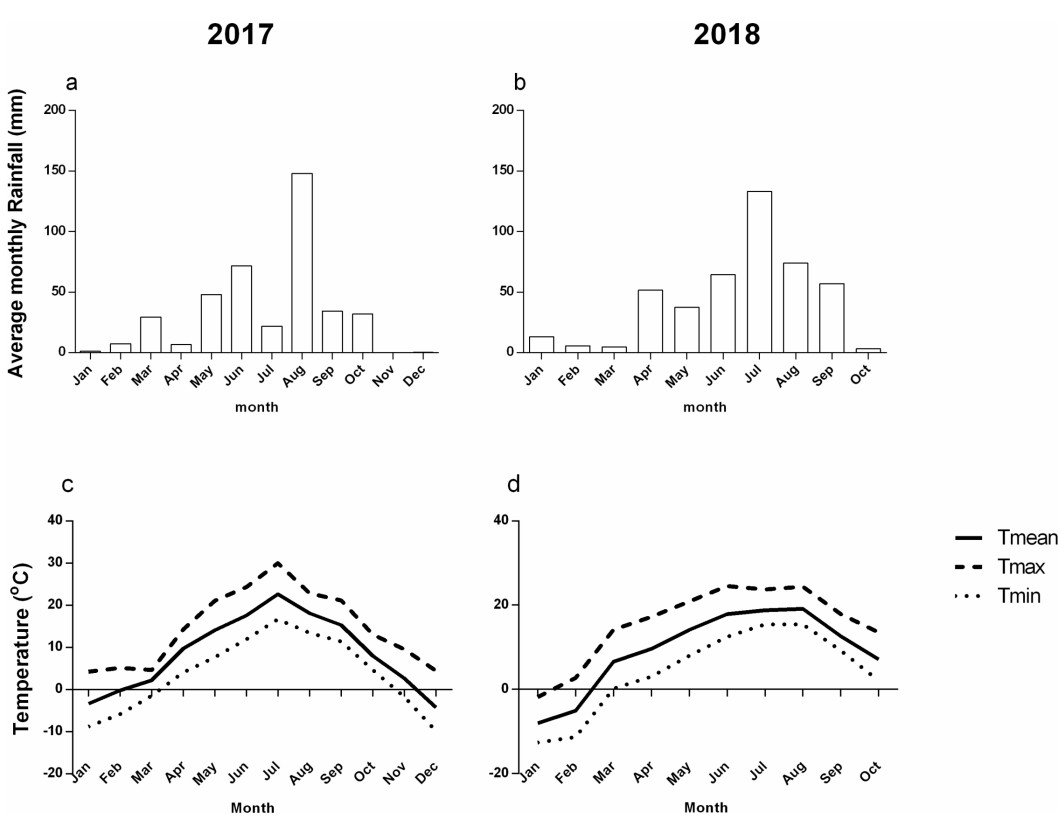

**Figure 1** Rainfall amounts for 2017 (A), 2018 (B) and mean, maximum and minimum temperatures for 2017 (C) and 2018 (D) in the Anjiapo catchment in Dingxi.

**Table 1  Properties of wheat straw mulch.**

| Parameter | Content (%) |
|---|---|
| Potassium | $0.54 \pm 0.05$ |
| Carbon | $40.19 \pm 3.2$ |
| Nitrogen | $0.81 \pm 0.1$ |
| Phosphorus | $0.09 \pm 0.01$ |

manually by the drill method in rows with row spacing of 25 cm while fertilizers were applied to all the plots using Di-ammonium phosphate ($N+P_2O_5$) at a rate of 146 kg/ha and urea (46%) at a rate of 63 kg/ha. Three rows per plot were harvested for determination of aboveground and below ground plant products at physiological maturity. Aboveground biomass was determined by oven drying of plants at 80 °C to constant weight (*Alhassan et al., 2018*), while grain yields were determined by oven-drying at 105 °C for 45 min (*Yeboah et al., 2016a*).

## Sampling and measurements
### Soil properties

Soil water content and soil temperature at 0–10 cm depth were measured using EM50 data logger and GS3 soil moisture, temperature and EC sensor (Decagon Devices, Inc., Pullman, Washington). The data was sampled every 2 min and subsequently downloaded onto the computer using the $ECH_2O$ software. Chamber temperature was recorded using a handheld digital thermometer (JM624, Jinming Instrument Co., Tianjing, China). Soil moisture and soil temperature data were taken concurrently with gas sampling.

Soils were sampled at 0–10, 10–20 and 20–40 cm with a soil auger (4 cm diameter) for determination of soil organic carbon (SOC) and soil total nitrogen (STN). SOC was determined by the Walkley-Black dichromate oxidation method (*Nelson & Sommers, 1982*) while STN was determined by the Kjeldahl digestion and distillation procedure as described by *Bremner & Mulvaney (1982)*.

## Gas sampling and Flux measurements

The gas sampling procedure was conducted between September, 2017 and January, 2019. The static dark chamber and Gas chromatography (GC) method as described by *Wang & Wang (2003)* were used for gas sampling and flux measurements. In each plot (a total of 12 plots), a stainless steel base with a collar ($50 \times 50 \times 10$ cm) was installed to support placement of the sampling chamber ($50 \times 50 \times 50$ cm) for gas sampling. Air samples were drawn from the chambers concurrently for the 3 replicates of each treatment. Samples were drawn at 5 different times at 0, 9, 18, 27, and 36 min respectively using 150 ml gas-tight polypropylene syringes and released into 100 ml aluminum foil sampling bags (Shanghai Sunrise Instrument Co. Ltd, Shanghai). Gas samples were then analyzed in the laboratory with a GC system (Echrom GC A90, China) equipped with a flame ionization detector (FID) for $CH_4$ and $CO_2$ analysis and Electron capture detector (ECD) for $N_2O$ analysis. The FID operates at a temperature of 250 °C, and $H_2$ flow rate of 35 $cm^3$ $min^{-1}$. Peak areas of $CO_2$, $CH_4$ and $N_2O$ were analyzed in Echrom-ChemLab software. Before the analyses of sample gases, calibrations were done with standard gas obtained from Shanghai Jiliang Standard Reference Gases Co., Ltd, China. Concentrations of the standard gases were 456.00 ppmv for $CO_2$, 2.00 ppmv for $CH_4$ and 0.355 ppmv for $N_2O$. The sample gas concentrations obtained for the five sampling times were plotted against time in order to obtain the change in concentration over the sampling time. $CO_2$ emissions in terms of ecosystem respiration ($R_{eco}$), $CH_4$ and $N_2O$ fluxes were calculated as shown in the File S1 following equation 1 (*Wei et al., 2014*). Further flux analysis, soil carbon input components, and global warming potential were calculated from equation 2 to 10 as shown in File S1 (*Bolinder et al., 2007*; *Zhang et al., 2009*; *Huang et al., 2007*; *IPCC, 2013*).

## Statistical analysis

The data was analyzed in SPSS, version 22 (IBM Corporation, Chicago, USA). One-way Anova was conducted and treatment means were separated using the Duncan's multiple range tests (DMRT) at $p < 0.05$. Linear and non-linear regressions were used to examine the relationships between crop yields, soil properties and greenhouse gas fluxes. The exponential

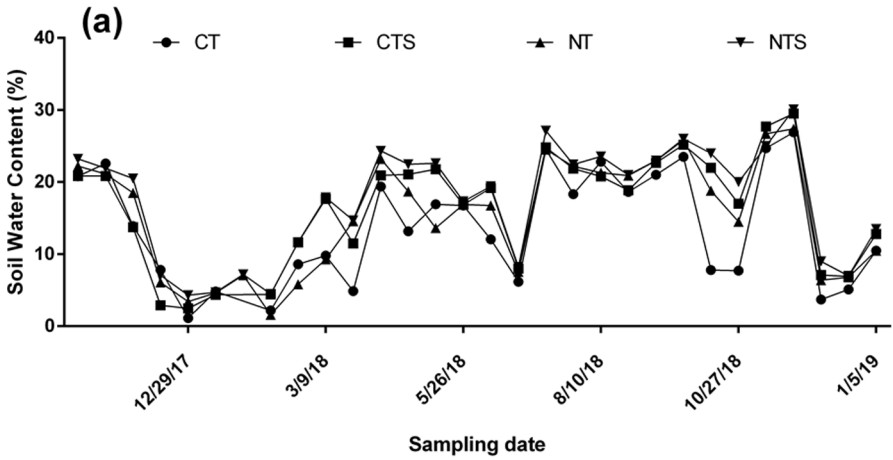

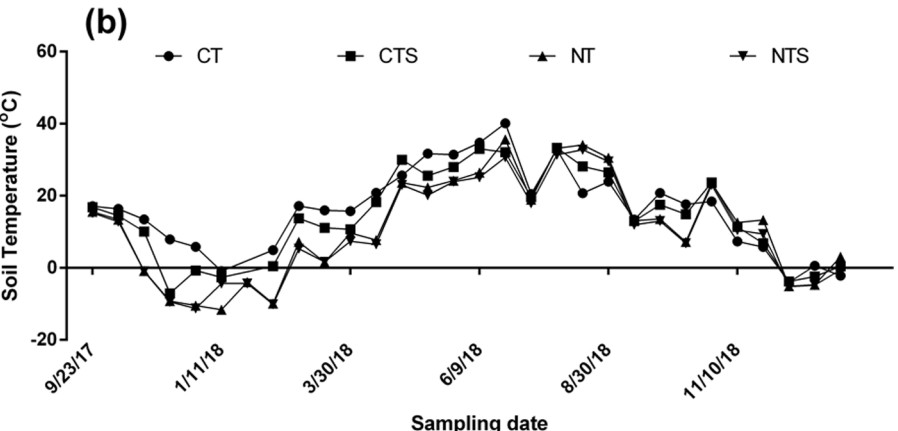

**Figure 2** Soil water content (A) and soil temperature (B) at various sampling times (10 cm depth).

and power equations were used to describe the relationship between ecosystem respiration, soil temperature and soil water content as shown in equations 11 and 12 respectively (File S1).

## RESULTS

### Soil water content and Soil temperature

Soil water content (SWC) was higher in NTS than all other treatments while CT had the lowest SWC at almost all sampling times (Fig. 2A). Conventional tillage with straw mulch (CTS) however stored more moisture than NT and CT at most sampling times. The highest SWC values were recorded in the growing season between July and September.

Soil temperature, as shown in Fig. 2B showed peak temperatures occurring in June-August. The highest temperatures were recorded in CT in most times while NTS and NT had the least temperatures in most instances.

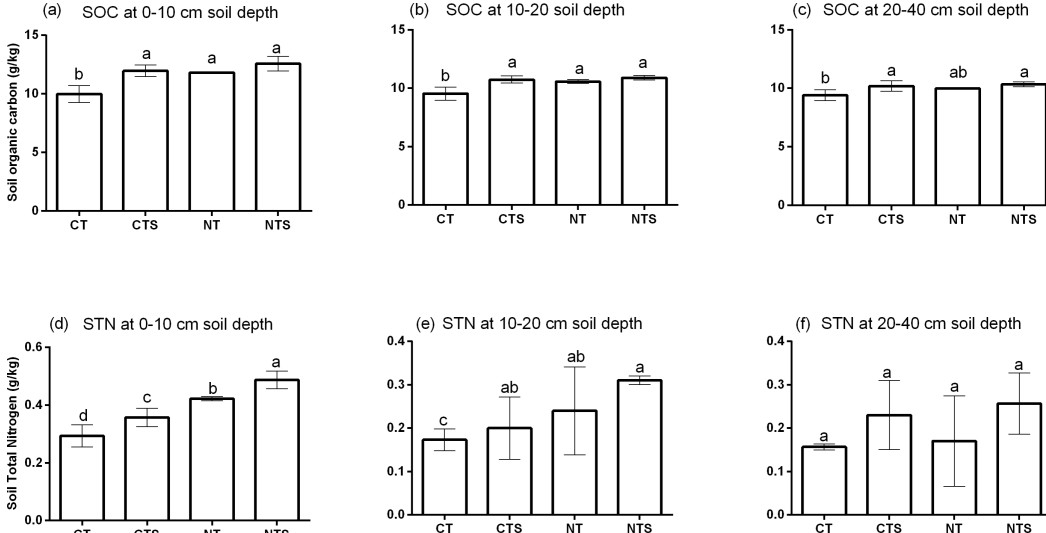

**Figure 3** **Soil organic carbon (SOC) and soil total nitrogen (STN) among tillage treatments within different depths.** (A-F) Treatments with common letters within a depth are not statistically different at $p \leq$ 0.05.

## Soil organic carbon (SOC) and soil total nitrogen (STN)

Conservation Agricultural practices increased SOC at all depths (Figs. 3A, 3B and 3C). At the 0–10 cm and 10–20 cm depths, SOC was increased significantly ($p < 0.05$) under CTS, NT and NTS. For the 0–10 cm SOC values were $9.98 \pm 0.73$, $11.97 \pm 0.5$, $11.81 \pm 0.09$ and $12.57 \pm 0.62$ gkg$^{-1}$ respectively for CT, CTS, NT and NTS. Compared with CT, SOC was increased by 19.95%, 18.38% and 26.03% respectively under CTS, NT and NTS within the 0–10 cm soil depth. A similar trend was observed within the 10–20 cm profile where SOC was also increased by 12.51%, 10.76% and 14.26% respectively under CTS, NT and NTS. However in the 20–40 cm depth there was a little deviation where SOC in NTS and CTS were significantly greater than CT but NT showed no significant difference. Meanwhile SOC decreased along soil depth irrespective of treatment.

There was also significance ($p <0.05$) in STN variations among treatments in the 0–10 cm and the 10–20 cm depths (Figs. 3D, 3E and 3F). At the 0–10 cm depth STN values were $0.29 \pm 0.04$, $0.36 \pm 0.03$, $0.42 \pm 0.01$ and $0.49 \pm 0.03$ gkg$^{-1}$ in CT, CTS, NT and NTS respectively. Compared with the control (CT), STN was increased significantly ($p<0.05$) by 21.59%, 43.75% and 65.91% respectively in CTS, NT and NTS. Similarly, within the 10–20 cm depth, STN was significantly ($p<0.05$) increased under CTS ($0.20 \pm 0.07$ gkg$^{-1}$), NT ($0.24 \pm 0.10$ gkg$^{-1}$) and NTS ($0.31 \pm 0.01$ gkg$^{-1}$) compared with CT ($0.17 \pm 0.01$ gkg$^{-1}$). Nonetheless there were no significant differences ($p >0.05$) in STN within the 20–40 cm depth. However there were observed reductions of STN along soil depth in all treatments.

## Grain yield

Tillage and straw treatments influenced grain yield in both years (Table 2). Average grain yields (2017-2018) were $644.61 \pm 116.40$, $854.46 \pm 76.51$, $699.30 \pm 133.52$ and $908.18 \pm$

**Table 2  Wheat grain yield response to different tillage treatments.**

| Treatment | 2017 Grain yield | 2018 | 2017-2018 |
|---|---|---|---|
| CT | $581.45 \pm 73.89^b$ | $707.78 \pm 96.49^b$ | $644.61 \pm 76.98^c$ |
| CTS | $587.69 \pm 35.96^b$ | $1121.23 \pm 54.19^a$ | $854.46 \pm 59.02^{ab}$ |
| NT | $653.36 \pm 27.25^b$ | $745.23 \pm 134.42^b$ | $699.30 \pm 64.88^{bc}$ |
| NTS | $854.46 \pm 25.33^a$ | $961.90 \pm 21.61^{ab}$ | $908.18 \pm 22.31^a$ |

38.64 kg ha$^{-1}$ respectively for CT, CTS, NT ad NTS. This means grain yield was increased by 32.55%, 8.48% and 40.89% respectively under CTS, NT and NTS compared with CT. CTS and NTS increased grain yield significantly ($p <0.05$) for the period (2017-2018) but grain yield was not significantly increased under NT. There were however slight interannual variations in yield response to treatments. Grain yields were generally higher in 2018 than in 2017. Also in 2017 NTS showed the highest grain yield but in 2018 CTS showed the highest grain yield. While only NTS significantly increased grain yield in 2017, in 2018 both NTS and CTS significantly improved grain yield.

## Average greenhouse gas emissions

Ecosystem Respiration for all treatments are shown in Figs. 4A and 4B respectively. Tilled soils emitted significantly more $CO_2$ than non-tilled soils. In the growing season, average $CO_2$ emission rates were $270.475 \pm 11.03, 262.88 \pm 0.20, 183.83 \pm 34.05$ and $190.72 \pm 19.20$ mg C m$^{-2}$ h$^{-1}$ in CT, CTS, NT, and NTS respectively, resulting in emission reduction in CTS, NT, and NTS by 2.81%, 32.03% and 29.48% respectively. In the non-growing season, emissions were relatively lower at rates of $30.55 \pm 1.71, 45.51 \pm 3.88, 31.74 \pm 1.35$ and $34.15 \pm 5.71$ mg C m$^{-2}$ h$^{-1}$ respectively in CT, CTS, NT, and NTS.

All the treatments served as minor sinks of $CH_4$ (Figs. 4C and 4D). The respective absorption rates were $-0.071 \pm 0.041, -0.102 \pm 0.005, -0.106 \pm 0.009$ and $-0.149 \pm 0.001$ mg C m$^{-2}$h$^{-1}$ for CT, CTS, NT and NTS in the growing season while in the non-growing season the values were $-0.081 \pm 0.064, -0.055 \pm 0.006, -0.071 \pm 0.018$ and $-0.055 \pm 0.004$ mg C m$^{-2}$ h$^{-1}$ respectively. However, there were variations in their sink capacities. NTS was the largest sink in the growing season while CT was the largest sink in the non-growing season. Generally, average absorption rates were higher in the growing season than in the non-growing season in all treatments except CT.

Averagely across seasons, all treatments served as emitters of $N_2O$, but flux values were statistically similar under all treatments in both seasons. Also, there were higher emissions in the growing season than the non-growing season (Figs. 4E and 4F). In the growing season, CTS had the highest emission of $N_2O$. The fluxes in the growing season were $3.09 \pm 1.96, 14.88 \pm 0.42, 11.39 \pm 6.80$ and $12.61 \pm 2.76$ µg N m$^{-2}$ h$^{-1}$ for CT, CTS, NT and NTS respectively while in the non-growing season, values of $N_2O$ fluxes ranged between 0.21 and 2.69 µg N m$^{-2}$ h$^{-1}$.

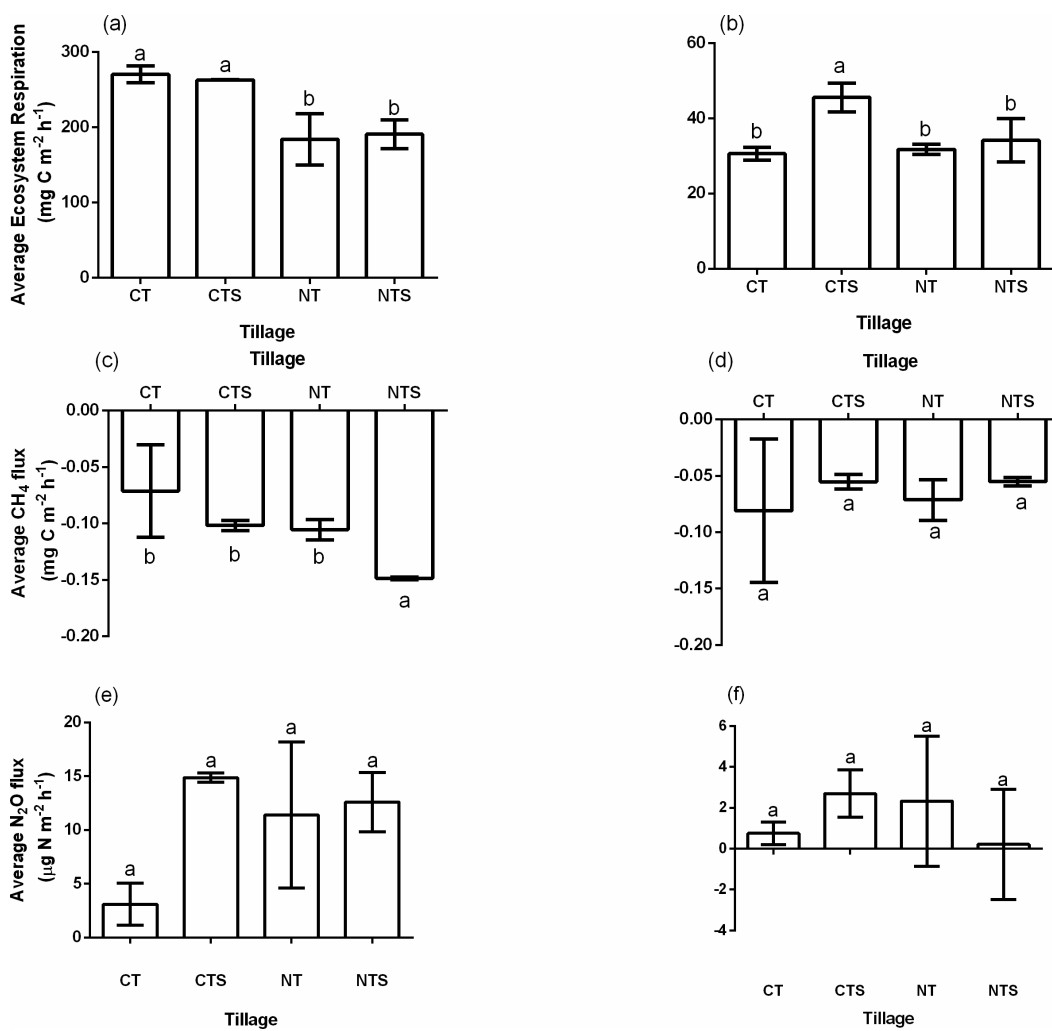

**Figure 4** Average ecosystem respiration, CH4 and N2O fluxes across treatments in growing season (A, C & E) and non-growing season (B, D & F). Error bars are standard errors, $n = 3$.

## Net GHG fluxes, Global warming potential (GWP) and greenhouse gas intensity (GHGI)

The Net $CO_2$-flux, $CO_2$ equivalents ($CO_2$e) of $CH_4$ and $N_2O$, GWP and GHGI of all treatments are shown in Table 3. Grain Yield (Table 1) and Harvest index were used to estimate the carbon components of harvest i.e., grain and straw in order to obtain Gross Primary production (GPP) and Net Primary Production (NPP). Harvest index, carbon in grain and straw, GPP and NPP are shown in File S2.

Net $CO_2$ fluxes in NTS and NT ($6.65 \pm 0.73$ and $7.37 \pm 0.89$ tCO$_2$e ha$^{-1}$y$^{-1}$respectively) were significantly lower than those in CT and CTS ($11.14 \pm 0.58$ and $10.65 \pm 0.18$ tCO$_2$e ha$^{-1}$y$^{-1}$respectively), showing reduced net carbon exchange into the atmosphere under NTS and NT. Similarly, GWP was greater in CT than all other treatments with significant reductions in NT and NTS ($p < 0.05$). Compared with CT, the reduction in GWP was

**Table 3  Net GHG fluxes, Global warming potential and Greenhouse gas intensity among tillage treatments.**

| | Net $CO_2$-flux (t$CO_2$e ha$^{-1}$y$^{-1}$) | $CH_4$ –$CO_2$e (t$CO_2$e ha$^{-1}$y$^{-1}$) | $N_2O$-$CO_2$e (t $CO_2$e ha$^{-1}$y$^{-1}$) | Net GWP (t$CO_2$e ha$^{-1}$y$^{-1}$) | GHGI (t$CO_2$e t$^{-1}$ grain) |
|---|---|---|---|---|---|
| **CT** | $11.14 \pm 0.58^a$ | $-0.21 \pm 0.017^b$ | $0.035 \pm 0.004^b$ | $10.96 \pm 0.56^a$ | $17.21 \pm 1.18^a$ |
| **CTS** | $10.65 \pm 0.18^a$ | $-0.23 \pm 0.005^{ab}$ | $0.22 \pm 0.016^a$ | $10.65 \pm 0.19^a$ | $12.56 \pm 0.9^{ab}$ |
| **NT** | $7.37 \pm 0.89^b$ | $-0.25 \pm 0.011^{ab}$ | $0.18 \pm 0.075^a$ | $7.30 \pm 0.97^b$ | $10.37 \pm 2.34^b$ |
| **NTS** | $6.65 \pm 0.73^b$ | $-0.27 \pm 0.024^a$ | $0.17 \pm 0.016^a$ | $6.55 \pm 0.70^b$ | $7.18 \pm 1.77^b$ |

**Notes.**
The sign convention adopted is positive (+) means emission whilst negative (−) means absorption.

**Table 4  Correlation between grain yield and soil chemical properties.**

| Soil chemical property | Grain Yield |
|---|---|
| **Soil Organic Carbon (SOC)** | |
| Soil organic carbon at 10 cm | 0.642[*] |
| Soil organic carbon at 20 cm | 0.614[*] |
| Soil organic carbon at 40 cm | 0.487 |
| **Total Nitrogen (TN)** | |
| Total nitrogen at 10 cm | 0.672[*] |
| Total nitrogen at 20 cm | 0.609[*] |
| Total nitrogen at 40 cm | 0.260 |

**Notes.**
[**]Correlation is significant at the 0.01 level (2-tailed).
[*]Correlation is significant at the 0.05 level (2-tailed).

2.83%, 33.40%, and 40.35% under CTS, NT and NTS respectively. The GHGI is a yield-scale quantification of GWP, thus is a factor of GWP and Grain yield. It shows the contribution of the cropping system to global warming per unit grain yield. Our results showed that CT was the highest contributor of global warming per unit grain produced compared with other treatments. Significant reductions of GHGI were found in NTS and NT ($p < 0.05$).

## Correlations between soil parameters and grain yield

Grain yields were highly influenced by soil nutrients. There were significant positive correlations ($p < 0.05$) of grain yield and SOC and STN (Table 4) at the 0–10 cm and 10–20 cm depths respectively (Table 4). However at deeper depths (20–40 cm), there were no significant correlations observed.

## Correlations between ST, SWC and greenhouse gases

Greenhouse gas fluxes were generally influenced by ST and SWC as shown in Table 5. $CO_2$ emission in the form of ecosystem respiration ($R_{eco}$) increased exponentially as ST increased; ST-$R_{eco}$ relationship followed an exponential function and was highly significant ($p < 0.001$) and positive with $R^2$ values of 0.68, 0.63, 0.80 and 0.80 respectively in CT, CTS, NT and NTS while SWC-$R_{eco}$ relationship was best described by a power function and also showed highly significant ($p < 0.001$) positive correlations with $R^2$ values of 0.06, 0.10, 0.14 and 0.07 respectively in CT, CTS, NT and NTS. Similarly, for $CH_4$ fluxes, ST-$CH_4$ relationship was an exponential function and highly significant as well ($p < 0.001$)

**Table 5  Correlation between greenhouse gases, soil temperature and soil water content.**

| Treatment | Soil temperature | | | Soil water content | | |
|---|---|---|---|---|---|---|
| | Equation | $R^2$ | $p$-value | Equation | $R^2$ | $p$-value |
| | | | **Ecosystem respiration** | | | |
| CT | $y = 33.04e^{0.07x}$ | 0.68 | <0.001 | $y = 363.57x^{0.48}$ | 0.056 | <0.01 |
| CTS | $y = 32.55e^{0.075}$ | 0.63 | <0.001 | $y = 474.18x^{0.67}$ | 0.095 | <0.001 |
| NT | $y = 21.58e^{0.08x}$ | 0.80 | <0.001 | $y = 483.20x^{0.88}$ | 0.138 | <0.001 |
| NTS | $y = 27.79e^{0.077x}$ | 0.80 | <0.001 | $y = 317.12x^{0.67}$ | 0.07 | <0.001 |
| | | | **$CH_4$ flux** | | | |
| CT | $y = -0.063e^{0.012x}$ | 0.019 | = 0.001 | $y = -0.07 - 0.05x$ | 0.03 | = 0.79 |
| CTS | $y = -0.052e^{0.026x}$ | 0.135 | <0.001 | $y = -0.08 - 0.02x$ | 0.04 | = 0.87 |
| NT | $y = -0.062e^{0.022x}$ | 0.085 | <0.001 | $y = 0.12 - 0.35x$ | 0.1 | = 0.05 |
| NTS | $y = -0.068e^{0.028x}$ | 0.174 | <0.001 | $y = -0.04 - 0.31x$ | 0.04 | = 0.137 |
| | | | **$N_2O$ flux** | | | |
| CT | $y = -0.84 + 0.12x$ | 0.016 | = 0.456 | $y = -2.45 + 26.62x$ | 0.01 | = 0.269 |
| CTS | $y = -1.77 + 0.66x$ | 0.209 | <0.01 | $y = -3.87 + 71.36x$ | 0.08 | = 0.075 |
| NT | $y = 4.64 + 0.12x$ | 0.003 | = 0.65 | $y = -0.68 + 43.45x$ | 0.09 | = 0.055 |
| NTS | $y = 2.05 + 0.28x$ | 0.08 | = 0.07 | $y = -3.6 + 48.92x$ | 0.08 | = 0.067 |

while SWC-$CH_4$ relationship was best described by a linear function, albeit not significant. Meanwhile ST and SWC did not seem to explain variations of $N_2O$ as the correlations were not significant except in ST-$N_2O$ relationship under CTS.

# DISCUSSION

Increased SWC and ST reduction in NTS (Fig. 2) is in line with other studies where conservation tillage improved SWC and soil water storage (*Lal et al., 2012*; *Li et al., 2014*). This could be attributed to the effect of straw (*He et al., 2011*; *Lal et al., 2012*). Straw reduced evaporation (*Kang et al., 2004*) leading to improvement in water retention (*Hill, Horton & Cruse, 1985*) and infiltration (*Li et al., 2011*). Straw mulch may also insulate the soil from direct impact of solar heat leading to decline of temperature.

Other studies have also reported increased SOC stocks after adoption of conservation tillage practices (*Ogle, Breidt & Paustian, 2005*; *Paustian et al., 2006*) which is similar to the findings of this study. Increased SOC and STN in CA plots could be attributed to less disturbance of soil which might reduce the risk of exposure of soil organic matter to decomposition process, thereby increasing SOC storage (*Lal, 2015*; *Reicosky, 1997*; *Six, Elliott & Paustian, 2000*). Also, favorable moisture content in CA plots (Fig. 2) may foster water and nutrient uptake by plant roots and also induce substrate movement for C fixation which may result in higher photosynthetic C input, leading to net C sequestration. Soil moisture and residue retention in CA plots may reduce wind and water erosion which could improve soil water storage and reduce leaching of soil nitrogen (*Allmaras & Dowdy, 1985*; *Lamb, Peterson & Fenster, 1985*).

Higher grain yield under CA practices (Table 1) is in tandem with other studies where conservation tillage increased grain yields (*Bordovsky, Choudhary & Gerard, 1998*;

*Halvorson et al., 2000*; *Li et al., 2014*; *Zheng et al., 2014*; *Yeboah et al., 2016a*; *Yeboah et al., 2016b*). This could be attributed to improved soil properties under these treatments. Higher SWC (Fig. 2) facilitated movement and uptake of available nutrients, as shown by higher SOC and STN stocks in NTS, NT and CTS (Fig. 3) thereby leading to higher grain yields. This plausibility is increased as Pearson correlation showed significant positive correlations between grain yield and these soil properties (Table 3).

Significant lower rates of ecosystem respiration ($p < 0.05$) in NT and NTS compared with the tilled soils (Fig. 4) was consistent with other studies (*Chaplot et al., 2012*; *Yeboah et al., 2016b*) where conservation tillage significantly reduced soil respiration. $CO_2$ emission rates is often controlled by a number of factors including: gradient of concentration of $CO_2$ between the atmosphere and the soil medium, soil water, soil temperature, wind speed and soil physical and chemical properties (*Raich & Schlesinger, 1992*). Tillage influences these factors directly and or indirectly which resultantly influences $CO_2$ emissions as well. Soil disturbance under conventional tillage, may trigger microbial activities and increase decomposition rates (*Al-Kaisi & Yin, 2005*), leading to higher $CO_2$ emissions. Soil disturbance may also increase soil aeration, resulting in higher oxidation of carbon into $CO_2$ (*Jackson et al., 2003*). On the contrary, under no-till, decomposition is slower due to less soil disturbance (*Curtin et al., 2000*). Higher soil temperature under CT (Fig. 2) may exponentially increase microbial activities (*Meixner, 2006*) while lower soil temperature may reduce microbial activity, hence reduce emissions in the conservation tillage plots (*Carbonell-Bojollo, De Torres & Rodríguez-Lizana, 2012*). This corroborates with the significant positive relationship between soil temperature and ecosystem respiration found in this study (Table 4).

All four tillage methods resulted in uptake of $CH_4$ in both growing and non-growing seasons. Other studies on the Loess Plateau obtained similar results (*Wan et al., 2009*; *Yeboah et al., 2016b*). *Shen et al. (2018)* indicated that agroecosystems in dry regions with minimal irrigation often act as $CH_4$ sinks due to aerobic soil conditions. This is due to oxidation of $CH_4$ under aerobic conditions (*Matson, Pennock & Bedard-Haughn, 2009*; *Schaufler et al., 2010*). Lower temperatures under NTS may have played significant role in high uptake of $CH_4$ in NTS. The dominant methanogen during high temperatures (Methanosarcinaceae) utilizes $H_2/CO_2$ and acetate as methane producing precursors, and produces far higher methane than the methanogen at lower temperatures (Methanosaetaceae), which uses only acetate as methane producing precursor (*Ding & Cai, 2003*).

Average $N_2O$ fluxes found in this study were in the range of fluxes reported by *Ma et al. (2013)* in their study of GHGs in a rice-wheat rotation under integrated crop management systems. Averagely, all treatments served as slight emitters of $N_2O$ (Fig. 4). This is also consistent with the study of *Yeboah et al. (2016b)* on the Loess Plateau. There was significant positive correlation between soil temperature and $N_2O$ emission in CTS. Higher temperatures and soil water content in the growing season where 70–80% of rainfall occurs (Fig. 1) may have triggered nitrification and denitrification processes (*Davidson & Swank, 1986*), leading to higher $N_2O$ emissions in this season. High rainfall may increase water filled pore space, which influences $N_2O$ emissions in agricultural

soils (*Dobbie & Smith, 2003*). *Trujillo-Tapia et al. (2008)* also reported positive correlation of $N_2O$ with soil temperature and soil water content. Higher emission in the growing season than in the non-growing season could also be related to fertilizer N application in the growing season and its interactive effect with wet conditions within this period on denitrification processes (*Cho, Burton & Chang, 1997*).

The GWPs (Table 2) were in the range as reported by *Ma et al. (2013)* but greater than those reported by *Yeboah et al. (2016b)*. Furthermore, GHGIs in this study were far higher than those found in other studies (*Qin et al., 2010*; *Ma et al., 2013*). Higher GWPs and GHGIs in this study could be attributed to a general lower grain yield (Table 1) which is typically associated with drylands. Lower grain yield may generally result in relatively low carbon input (File S2) which may in turn result in relatively higher net GHG emissions. However GWP being lower under NT and NTS than in CT and CTS is attributable to relatively higher carbon input from the above ground plant product coupled with lower ecosystem respiration and higher $CH_4$ uptake in these plots.

## CONCLUSIONS

This study hypothesized that no-till and the application of straw improved soil chemical and physical properties, increased crop yield and reduced greenhouse gas emissions by comparing three conservation practices (conventional tillage with straw (CTS), No-till (NT) and no-till with straw (NTS)) to conventional tillage (CT). Our study showed that conservation tillage practices especially NTS improved soil water content and reduced soil temperature. Soil organic carbon and total nitrogen were also significantly improved under conservation practices especially within the top soil layer (0–20 cm). There was also significant improvement in average grain yield under NTS and CTS. Conservation tillage further reduced net $CO_2$ flux; increased $CH_4$ absorption but only slightly influenced $N_2O$ emissions in the dryland ecosystem. NTS and NT significantly reduced GWP and yield-scale GWP. For sustainability of arid and semi-arid cropping systems and for environmental quality, we recommend the adoption of conservation agricultural practices such as no-till, straw mulch, green manuring, contour ploughing and terracing on the Loess Plateau. Furthermore crop genetic and breeding techniques such as the use of drought resistant crop varieties should also be explored in order to enhance climate change resilience of crop production in dryland areas and reduce climate footprint of these areas.

### Funding

This research was supported financially by the National Natural Science Foundation of China (31560378, 31560343) and the Gansu Key Research and Development Program (20YF8NA135). The funders had no role in study design, data collection and analysis, decision to publish, or preparation of the manuscript.

### Grant Disclosures

The following grant information was disclosed by the authors:

National Natural Science Foundation of China: 31560378, 31560343.
Gansu Key Research and Development Program: 20YF8NA135.

## Competing Interests

The authors declare there are no competing interests.

## Author Contributions

- Abdul-Rauf Malimanga Alhassan and Chuanjie Yang conceived and designed the experiments, performed the experiments, analyzed the data, prepared figures and/or tables, authored or reviewed drafts of the paper, and approved the final draft.
- Weiwei Ma analyzed the data, authored or reviewed drafts of the paper, and approved the final draft.
- Guang Li conceived and designed the experiments, performed the experiments, authored or reviewed drafts of the paper, and approved the final draft.

## Data Availability

Data are available at Figshare:

Alhassan, Abdul-Rauf; Yang, Chuangjie; Ma, Weiwei; Li, Guang (2019): DATA_Influence of conservation tillage on Greenhouse gases fluxes and crop productivity in spring-wheat agroecosystems on the Loess Plateau of China. figshare. Dataset. https://doi.org/10.6084/m9.figshare.9923975.v2.

## Supplemental Information

Supplemental information for this article can be found online at http://dx.doi.org/10.7717/peerj.11064#supplemental-information.

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
