# Peer review of "Influence of conservation tillage on Greenhouse gas fluxes and crop productivity in spring-wheat agroecosystems on the Loess Plateau of China"

_PeerJ, doi:10.7717/peerj.11064_

## Round 0.1 · original submission · Major Revisions

Although the reviewers found the scientific values of the present research, there are major concerns, especially from the first reviewer, such as that the English should be improved and the research question was not explicitly stated. Please address the reviewers' comments in the revision and provide an itemized response to each of them.

Reviewer 1 ·

Basic reporting

- The English language needs to be improved throughout the manuscript. The literature is referenced properly but I suggested to use information based citation instead of the author (for e.g., There is an improved in Carbon sequestration (Smith et al., 2008)).
- The knowledge gap was not explained well.
- There are lots of figures so the author should emphasis on main figures that explained the story well.

Experimental design

The research article aligns with the scope of the journal. As a reader, I felt that the research question was not defined well although the gap is mentioned but not explained well. The methodology section needs to be revised thoroughly. A lot of detail was presented in the methodology section which is not needed. So, I would urge the author to trim this section. A special section for the equations is also needed.

Validity of the findings

The way the result is presented, the author had provided detailed fieldwork with an extensive dataset over the year 2017-2018. The author also provides an extensive literature review on the discussion. I would recommend to make a list of the important take-home messages from the Result as well as the Discussion section and then described that. Because the sections are too long and I am lost as I read through the paragraph. Please revised the section as the edits provided in the attachment.

Additional comments

This paper needs to highlight the why conservation tillage no-till had influence on the gas fluxes and crop productivity. Every section is lengthy so need to trim it. The author should pick up the main point and explained on it. All underlying data have been provided they are robust, statistically sound, &controlled.

Author needs to present a more comprehensive view of the literature surrounding the use of no-till on the gas fluxes and crop productivity. I recommend that the author works on each feedback the reviewer provided to make this paper better. Major revisions are necessary before publication.

Major comments:

Each section is long. The author makes sure to work on the grammar and concise the paper. It is very long. Some of the sections from the material and methodology can be combined together for e.g., Experimental design and this section can be combined together. Even for the result section, some of the paragraphs can be combined together. The discussion section is too long and I would suggest the author get the main take-home message in the section. The author should pick up the important figures and tables in the manuscript. Please look at the edits on the attachment.
The attachment contains specific comments and edits.
Line 42: What is the difference between food systems and Agriculture?
Line 43: You can put these lines on your introductory sentence is why agriculture accounts for the global warming effects?
Line 48-54: The author can list the advantages of cropping systems on carbon sequestration as the value (cite).
Line 58-61: soil physical properties (cite), chemical properties (cite) and biological properties (cite)
Line 61-63: This can be added after line 61 to show the pieces of evidence from the past studies of why conservation no-till is important. I would recommend the author to rewrite this paragraph especially from lines 64-74.
Line 75: What is radiative foraging?
Line 83: Thus the objectives of this study are:
Line 94-97: For this study, we have contentious data of 42 years (Put the weather parameter inside).
Line 198-200: Please revised this sentence.
A separate supplemental file including the equations should be useful for both the authors and the readers.
Line 254: Especially on the result section, the author need to rewrite this section. I am lost as I read through each section. Bring the most notable result on these sections. Please organize your result section by the importance of the issues.

Annotated reviews are not available for download in order to protect the identity of reviewers who chose to remain anonymous.

Reviewer 2 ·

Basic reporting

Overall, it was a nice scientific article which was meaningful for developing sustaintable and climate-smart agriculture. Besides, this article had clear structure and rich content with excellent writing. However, some parts of this article need to be improved:

1.Why most of your reference was Font Bold in author’s name and published year, but some didn’t, please make sure the demands of journal and unify it.

2.More detail about your research object (Loess Plateau and spring-wheat) need to be introduced in the introduction, e.g. why you focus on this, what is the characters of this.

3.Your abstract needs more detail about the meaning of the research, especially in the first sentence, while also make the results more concise.

4.Why the majority of your figure and table were black and white, while the Fig. 6&7&8 were colorful, using uniform color might be better.

5.It would be easier to read if you could arrange the soil properties measurements together (e.g. SOC, TN, Soil water content).

Experimental design

1.Your experimental design and writting about materials and method were nice. However,it would be better if you could provide more data about soil properties in line 97(not only SOC and TN), and in line 114, what kind of weight of wheat straw, wet or dry? And it would be better if you could provide the chemical properties of straw (C%, N%...).

2. If you could give the local factor(chinese) in line194&195 instead of Canadian, the results would be more convincing.

Validity of the findings

This paper verified the hypothesis well and explained the scientific problem adequately. 1.However,explaining the GHGs emission trends with more detail about weather data (rain fall and tempetatures) would be better for your discussion to illustrate how it change.

2.For Fig.7, have you tried to regress the data with all observations, not only the average value of 4 treatments.

3.Could you please explain that why large error bar was existed in CT treatment of CH4 in Fig.4, while others was small.

Additional comments

The experimental design, writing ability and innovation level of this article are all excellent, but some details need to be improved. Overall, it was a very good scientific paper.

---

## Round 0.2 · Minor Revisions

Both reviewers have found the manuscript has been greatly improved. However, there are still some problems to be fixed . Please rephrase the abstract, and improve the presentations of result and discussion sections. Respond to other comments as well.

Reviewer 1 ·

Basic reporting

Abstract still needs to be improved. The introduction is improved a lot from the last version. Delete the line 41 first sentence.

Experimental design

No Comment.

Validity of the findings

Some of the discussion section as line 280-282; 303-305 and 335-337 should be moved to result section.

Additional comments

Its improved but I still felt the result , discussion and the conclusion can be made better.

Reviewer 2 ·

Basic reporting

1. The problem about reference has been solved. Good.
2. The relevant information has been supplemented. Good.
3. The graphic and table were tidy.Good.
4. The expression of soil properties has been optimized. Good

Experimental design

In this part, my problems has been solved and the suggestions were also adopted. Good.

Validity of the findings

All problems I mentioned last time has been solved. Only one suggestion is that I hope the author could add more discussion about the uncertainties about GHGs emission in the article.

Additional comments

The design of the experiment and writing of this article are pretty good. Overall, it was a nice scientific paper.And it would be good if more suggestion about mitigating GHGs for the local situation.

---

## Round 0.3 · accepted · Accept

All reviewers have agreed the manuscript has the quality to be published on PeerJ. Please make sure to read the reviewer's last comment in the attached file and make a final round of correction before finalizing the paper.

Reviewer 1 ·

Basic reporting

Clearly written

Experimental design

Research questions well defined.

Validity of the findings

Conclusions are well stated, linked to original research question & limited to supporting results.

Additional comments

There was only one place where I suggested the author add some literature review. Otherwise, the manuscript has significantly improved over time.

Annotated reviews are not available for download in order to protect the identity of reviewers who chose to remain anonymous.